# Uncovering the Relationship between Selenium Status, Age, Health, and Dietary Habits: Insights from a Large Population Study including Nonagenarian Offspring from the MARK-AGE Project

**DOI:** 10.3390/nu15092182

**Published:** 2023-05-04

**Authors:** Robertina Giacconi, Francesco Piacenza, Valentina Aversano, Michele Zampieri, Alexander Bürkle, María Moreno Villanueva, Martijn E. T. Dollé, Eugène Jansen, Tilman Grune, Efstathios S. Gonos, Claudio Franceschi, Miriam Capri, Birgit Weinberger, Ewa Sikora, Olivier Toussaint, Florence Debacq-Chainiaux, Wolfgang Stuetz, Pieternella Eline Slagboom, Jürgen Bernhardt, Maria Luisa Fernández-Sánchez, Mauro Provinciali, Marco Malavolta

**Affiliations:** 1Advanced Technology Center for Aging Research, IRCCS INRCA, 60121 Ancona, Italy; 2Department of Experimental Medicine, Sapienza University of Rome, 00161 Rome, Italy; 3Molecular Toxicology Group, Department of Biology, University of Konstanz, P.O. Box 628, 78457 Konstanz, Germany; 4Human Performance Research Centre, Department of Sport Science, Universityof Konstanz, P.O. Box 30, 78457 Konstanz, Germany; 5Centre for Health Protection, National Institute for Public Health and the Environment, P.O. Box 1, 3720 BA Bilthoven, The Netherlands; 6Department of Molecular Toxicology, German Institute of Human Nutrition Potsdam-Rehbruecke (DIfE), 14558 Nuthetal, Germany; 7NutriAct-Competence Cluster Nutrition Research Berlin-Potsdam, 14458 Nuthetal, Germany; 8National Hellenic Research Foundation, Institute of Biology, Medicinal Chemistry and Biotechnology, 11635 Athens, Greece; 9Department of Medical and Surgical Sciences, University of Bologna, 40126 Bologna, Italy; 10Laboratory of Systems Medicine of Healthy Aging, Institute of Biology and Biomedicine and Institute of Information Technology, Mathematics and Mechanics, Department of Applied Mathematics, Lobachevsky State University, 603105 Nizhny Novgorod, Russia; 11Interdepartmental Center—Alma Mater Research Institute on Global Challenges and Climate Change, University of Bologna, 40126 Bologna, Italy; 12Institute for Biomedical Aging Research, Universität Innsbruck, 6020 Innsbruck, Austria; 13Laboratory of the Molecular Bases of Ageing, Nencki Institute of Experimental Biology, Polish Academy of Sciences, 3 Pasteur Street, 02-093 Warsaw, Poland; 14URBC-NARILIS, University of Namur, Rue de Bruxelles, 61, 5000 Namur, Belgium; 15Institute of Nutritional Sciences, Department of Food Biofunctionality, University of Hohenheim, 70593 Stuttgart, Germany; 16Department of Molecular Epidemiology, Leiden University Medical Centre, 2333 ZA Leiden, The Netherlands; 17BioTeSys GmbH, Schelztorstr. 54–56, 73728 Esslingen, Germany; 18Department of Physical and Analytical Chemistry, Faculty of Chemistry, University of Oviedo, Julian Clavería, 8, 33006 Oviedo, Spain

**Keywords:** selenium fractionation, plasma selenium, longevity, aging, inflammation

## Abstract

An inadequate selenium (Se) status can accelerate the aging process, increasing the vulnerability to age-related diseases. The study aimed to investigate plasma Se and Se species in a large population, including 2200 older adults from the general population (RASIG), 514 nonagenarian offspring (GO), and 293 GO Spouses (SGO). Plasma Se levels in women exhibit an inverted U-shaped pattern, increasing with age until the post-menopausal period and then declining. Conversely, men exhibit a linear decline in plasma Se levels with age. Subjects from Finland had the highest plasma Se values, while those from Poland had the lowest ones. Plasma Se was influenced by fish and vitamin consumption, but there were no significant differences between RASIG, GO, and SGO. Plasma Se was positively associated with albumin, HDL, total cholesterol, fibrinogen, and triglycerides and negatively associated with homocysteine. Fractionation analysis showed that Se distribution among plasma selenoproteins is affected by age, glucometabolic and inflammatory factors, and being GO or SGO. These findings show that sex-specific, nutritional, and inflammatory factors play a crucial role in the regulation of Se plasma levels throughout the aging process and that the shared environment of GO and SGO plays a role in their distinctive Se fractionation.

## 1. Introduction

Selenium (Se) is an essential element for human health, known for its antioxidant and immune-modulatory properties and for being the unique trace element incorporated into selenoproteins in the form of selenocysteine [1]. Se deficiency has been implicated in many conditions involving oxidative stress and inflammation, including cardiovascular diseases and cancer [2,3,4,5]. The Recommended Dietary Allowance (RDA) of Se for people above 50 years was reported around 55 µg, both in males and in females [6]. This amount of dietary Se is sufficient to promote physiological protection against oxidative stress, mainly through selenoproteins, and to reduce the risk of major pathologies associated with aging. Some studies have provided evidence for an association of circulating levels of serum Se and its antioxidant action with longevity [7,8,9,10], but the reference ranges are not well established in view of the large variations in Se status between countries [11]. Indeed, diet and the geographical area play an essential role in the presence of Se in human plasma, as demonstrated by studies comparing serum Se levels in people from areas characterized by different concentration of Se in the soil [12]. When the soil is rich in Se, the main food sources of Se (fish and seafood, cereals, whole meal bread, cereals, and white meat) contain higher levels of this trace element [13].

The total amount of Se in plasma is very important, but its subspecies are responsible for the biological activity [14,15]. The human selenoproteome contains 25 known selenoproteins (proteins that include at least a selenocysteine) [16]. Among these, selenoprotein P (SELENOP) and plasma glutathione peroxidase (GPX) are the most abundant species [17], and they play a relevant role for health status for the prevention of inflammation, cardiovascular diseases, infections, and cancer [18]. A relatively high amount was also found for the selenocompound Selenoalbumin (SeAlb), a selenium species that does not include a selenocysteine. To date, few studies have been conducted on the distribution of Se species in human blood and their correlation with overall Se levels. Moreover, while there is evidence on the decrease in circulating Se in nonagenarians and centenarians [19], no investigation has been carried out on Se species in relation to longevity.

In order to fill this gap, Se speciation in large multicenter studies focused on aging is desirable. Speciation of Se is generally performed with accurate but sophisticated procedures [20,21,22], which may prevent their usage in large multicenter studies that include thousands of subjects and limited amounts of plasma for the analysis. However, this problem has been recently overcome using micro separation techniques coupled with inductively coupled plasma mass spectrometry (ICP-MS) [23]. In the present study, we applied this procedure to investigate plasma Se and Se fractions on a large scale of subjects, including community-dwelling older adults (Randomly recruited Age-Stratified Individuals from the General population [RASIG]), GO (offspring of nonagenarians), and spouses of GO (SGO) recruited in the framework of the European MARK-AGE project [24,25].

## 2. Materials and Methods

### 2.1. Study Population, Recruitment, Data and Blood Collection 

MARK-AGE is a European-wide cross-sectional population study aimed at the identification of biomarkers of aging [24,25]. In the present work, we used plasma samples from a total of 3007 donors in the age range of 35–75 years recruited in eight different European countries.

Details of the recruitment procedures and of the collection of anthropometric, clinical, and demographic data have been previously published [26,27]. The plasma isolation procedure from blood, as well as the shipment and distribution of biological samples, have been described [26]. Briefly, lithium heparin plasma was prepared from whole blood, obtained using phlebotomy after overnight fasting, and subsequently stored at −80 °C. Samples were then shipped from the various recruitment centers to the MARK-AGE Biobank located at the University of Hohenheim, Stuttgart, Germany. From the Biobank, coded samples were subsequently sent to the IRCCS INRCA on dry ice, where they were stored at −80°C until use.

### 2.2. Determination of Se in Plasma and Fractionation Analysis

Plasma Se levels were determined using a Thermo XII Series ICP–MS device (Thermo Electron, Waltham, MA, USA) as previously reported [23]. Plasma samples were centrifuged at 20,000× *g* for 10 min, and the supernatants were diluted 1:20 (total volume 1 mL) with a diluent containing 0.1% Triton X-100 (BDH Chemicals), 0.1% Trace Select Ultra HNO3 (Sigma–Aldrich), and 10 ppb Rh (Merck) as the internal standard. External multielement calibration solutions containing Fe, Zn, Cu, and Se (blank and 0.5–500 ppb) were prepared via serial dilution of a parent multielement solution (Inorganic Ventures) using the same diluent used for the samples. The ICP-MS was set by using Collision cell technology (CCT) with kinetic energy discrimination (KED) with a gas mixture containing 8% H_2_ in He in order to avoid polyatomic interferences. Data (three repeats per sample) were acquired for ^80^Se.

The fractionation analysis was performed by coupling anion exchange HPLC to ICP–MS following the procedure already published by the authors in 2012 [23]. Briefly, plasma samples were injected directly into a monolithic anion exchange micro column (Dionex ProSwift SAX1-S, 1x50 mm i.d.). The buffers used for the chromatographic separation were 10 mM Tris (pH 7.4) (buffer A) and 1 M ammonium acetate in 10 mM Tris (pH 7.4) (buffer B). The HPLC separation of metal fractions was optimized at a flow rate of 0.2 mL/min, linearly increasing the concentration of buffer B from 0% (100% buffer A) to 60% (40% buffer A) in 10 min (“elution program”). A tandem LC mode was used to save the time needed for washing and regenerating the columns after each run. In tandem LC mode, two identical columns are switched between two flow paths: an analysis flow path and a regeneration flow path to allow column washing and re-equilibration off-line. While one column was equilibrated, the system injected the next sample into the other. The program used to wash and regenerate the columns off-line was the following: 60% B to 100% B in 0.5 min, hold 4 min, return to 0% B in 0.5 min, and equilibrate with 0% B for 5 min. The interface between the LC system and the ICP–MS device for automated run and acquisition of chromatograms was performed with an external trigger card included in an LC coupling kit (Thermo Fischer).

The ICP-MS was set by using CCT and the same gas mixture used for Se plasma determination. Data were acquired for ^78^Se. The main fraction of Se (Se1) eluted after 2.5 min, with a second small peak (Se2) eluting at 5.3 min. Se1 and Se2 areas were defined following the retention times already considered in the methodological manuscript [23]. The relative percentage of the areas under the peaks was used for the analysis.

### 2.3. Statistical Analysis

Characteristics of the population studied were described using means and SD for continuous variables (age, BMI) and frequencies (%) for the categorical variables (sex, BMI classes, age class, groups, and country). The identification of potential critical variables that can affect the levels of Se was performed via non-parametric test (test of Kruskal–Wallis or Mann–Whitney U for two comparisons of the group) and Generalized Linear Models (GLMs) including age, gender, BMI, and country effects. Se plasma levels did not follow a normal distribution (Kolgomorov–Smirnov test) considering the whole population, but the data displayed a normal distribution when considered within age classes and country, particularly after log transformation. Critical variables identified by non-parametric tests and GLMs were subsequently included as covariates in additional GLM models to establish the influence of major confounders on target variables (age classes and subject groups). For this analysis, log transformation was applied to all continuous variables included in the analysis. Pairwise comparisons (Bonferroni’s method for GLM tests) were used to identify significant differences between percentile groups of each categorized variable. Linear regression analysis among plasma Se levels and other continuous variables was performed with a bootstrap procedure (1000 samples) stratified by the major categorical factors affecting the target variables. Multivariate stepwise linear regression analysis was also performed to investigate the association among plasma Se levels with glucometabolic, inflammatory, and anti-inflammatory parameters. The variables included were age, sex, country, food consumption, smoking habit, alcohol consumption, body mass index (BMI), lipid profile, albumin, glucometabolic parameters (fasting blood glucose, glycosylated hemoglobin, insulin), inflammatory parameters (C-reactive protein, fibrinogen, homocysteine, ceruloplasmin), and anti-inflammatory parameters (adiponectin). All statistical analysis (excluding analysis of batch effects) was carried out using SPSS software (SPSS Inc., Chicago, IL, USA; Version 23.0).

## 3. Results

### 3.1. Characteristics of the Study Population

The analysis of plasma Se levels was performed on samples obtained from donors (3007 individuals) from eight European countries. Characteristics of the study population are summarized in Appendix A. A total of 2200 subjects were from the RASIG subpopulation, representing individuals from the general population. A total of 514 subjects were from the nonagenarian offspring subpopulation called GO (“GEHA offspring “). These subjects had a long-lived parent who had previously been enrolled in the project GEHA [28]. GO are considered as a potential model of “delayed aging”, based on the assumption that their genetic background may predispose them to longevity. A total of 293 spouses of GO (SGO) [24,25] were included as a control for environmental factors and lifestyle. The MARK-AGE subjects covered the age range between 35 and 75 years and were stratified into four 10-year age groups. The age distribution of RASIG individuals was almost homogeneous, while GO and SGO individuals fell into age ranges greater than 54 years. The distribution of GO, SGO, and RASIG among the recruiting centers is reported in Appendix A.

The composition of males and females was mainly comparable in all age groups. In accordance with a previous investigation on the MARK-AGE population [29] and with the literature data, the body mass index (BMI) increased with age from 35 to 75 years, indicating that the analyzed population was effectively representative of a physiological aging process (Appendix A). The characteristics of the study population referred to each recruiting center were similar to previous published data [29].

### 3.2. Levels of Plasma Se Concentrations According to Age and Demographics

In the RASIG plasma, Se displays a non-normal distribution across the whole population (by Kolmogorov–Smirnov test and Shapiro–Wilk test). However, when we performed the test of normality separating the dependent variables into subgroups (considering center, age group, and experimental groups), all variables showed a normal distribution in the majority of subgroups, with few exceptions. This can be also seen from the similar values of the means reported in Table 1 and the medians reported in Appendix A. Nevertheless, we used both parametric (e.g., Generalized Linear Models) (Table 2) and non-parametric analyses (e.g., Kruskal–Wallis in Appendix A) to confirm the results.

The levels of plasma Se in RASIG display modest changes with age, best modelled with a non-linear pattern (Appendix A), with a slight increase in the 55–64-year class compared to the youngest and oldest age classes (Table 1). However, sex stratification revealed that Se plasma levels decreased linearly with age in males (coefficient: −0.086, *p* < 0.05, by generalized linear mixed model adjusted for BMI, country, fish and vitamin consumption), while females showed a U-shape trend (Figure 1). Males older than 65 years had lower Se plasma levels than subjects in the 35–44 age group (*p* < 0.05).

Females in the 35–44 age group had lower Se plasma levels than those in the 45–54 and 55–64 age groups (*p* < 0.0001), while old females over 65 showed reduced Se plasma levels compared to those in the 45–54 (*p* < 0.05) and 55–64 (*p* = 0.001) age groups. Due to this different pattern, sex differences were observed in the age class 35–44 (males higher than females, *p* < 0.001) and in the age class 55–64 (females higher than males, *p* < 0.001), not taking into account the whole population (Table 1).

Recruitment center had a strong effect on plasma Se, with the highest levels observed in Finland and the lowest levels in Poland. Plasma Se levels were inversely related to BMI, with the lowest levels observed in the ≥30 BMI class (Table 1).

### 3.3. Influence of Dietary and Inherited Factors on Plasma Se Levels

Since plasma Se levels are expected to change according to dietary habits, we carried out an automated regression analysis of plasma Se including fruit, vitamin, vegetables, bread, meat, fish, eggs, and dairy products consumption. According to this exploratory analysis, the two major dietary factors affecting plasma Se levels were fish and vitamin consumption (Appendix A).

Hence, we included these factors and the subject groups (GO, SGO, RASIG) in a multivariate analysis of plasma Se, performed on the whole population with the aim to investigate the respective influence of the inherited “longevity background” and dietary habits on these variables (Appendix A). Plasma Se levels were not different between the RASIG, GO, and SGO groups, but they were influenced by dietary habits (*p* < 0.001). Plasma Se levels, in relation to vitamin and fish consumption and subject groups, are also represented in Figure 2, which shows an increment of this metal trace element in RASIG and GO subjects who take more vitamins or consume more fish. In Appendix A, plasma Se levels in the whole population in relation to vitamin and fish consumption after country stratification are also displayed.

### 3.4. Selenium Fractions in MARK-AGE Population

The % distribution of Se, among the fractions identified in plasma by our HPLC system, was performed following the nomenclature previously assigned without identification of the representative selenoproteins [23]. The method can separate three peaks, specifically the Se1a, Se1b, and Se2 peaks. The latter is mirrored in Se1a + Se1b (as their sum is 100%).

In the RASIG population, Generalized Linear Models analysis showed that younger females had lower Se1a percentages than those in the 45–54 and 55–64 age groups (*p* < 0.01), while Se1a in males did not change in relation to age groups (Appendix A). Se1b was lower in younger females compared to those in the 65–75 age class (*p* < 0.05). No significant differences were found in males (Appendix A). Moreover, after sex stratification, Se2 decreased linearly with age in females (coefficient: −0.058, *p* < 0.05, by generalized linear mixed model adjusted for BMI, country, fish and vitamin consumption), while males showed an inverted U-shape trend (Figure 3A). 

Females in the 35–44 age group had lower Se2 than those in the 55–64 (*p* < 0.01) and 65–75 (*p* < 0.05) age groups. Males in the 45–54 age group had higher Se2 than subjects in the 35–44 and 55–64 age groups (*p* < 0.05). 

Se1b and Se2 percentages significantly differ between subject groups, independent of nutritional or other confounding factors (generalized linear mixed model adjusted for age, gender, country, BMI, fish and vitamin consumption). Se2 was higher in RASIG compared to GO and SGO (Figure 3B), while after country stratification, Se2 was augmented in RASIG compared to GO in Italy, Greece, and Belgium (Appendix A). Se1b was significantly lower in RASIG than in GO and SGO subjects (*p* < 0.05), while Se1a was similar between subject groups (Appendix A). In Appendix A is depicted a representative comparison between RASIG (red line) and GO (black line) Se chromatographic separations.

### 3.5. Association of Glucometabolic, Inflammatory, and Anti-Inflammatory Factors with Plasma Se Levels and Se2 Fraction

Independent glucometabolic and inflammatory markers associated with plasma Se levels and Se2 fraction (dependent variable) were identified with a linear regression analysis using the stepwise method (Table 2). We observed a positive association for Se plasma levels with albumin levels (*p* < 0.0001), HDL total cholesterol, fibrinogen (*p* < 0.01), and triglycerides (*p* < 0.05) and a negative association with homocysteine (*p* < 0.05). No association was found with circulating concentrations of fasting blood glucose, glycosylated hemoglobin (HbA1c), C-reactive protein, ceruloplasmin, and adiponectin.

Se2 was negatively associated with fibrinogen (*p* < 0.0001) and positively associated with adiponectin (*p* < 0.0001), H1Abc (*p* < 0.01), and ceruloplasmin (*p* < 0.01).

## 4. Discussion

In the present work, we evaluated plasma Se levels and individual Se fractions in relation to aging and longevity, considering the effect of countries and dietary habits, as well as the association of plasma Se concentrations and Se2 fraction with glucometabolic, inflammatory, and anti-inflammatory factors. Plasma Se varied greatly in relation to the country of residence, and subjects from Finland in particular showed the highest plasma Se, while those from Poland had the lowest values. The relatively low levels of Se observed in the other countries (Greece, Poland, and Austria) might suggest the use of Se biofortification to produce Se-enriched foods for counteracting Se deficiency. Consumption of Se-biofortified wheat or maize flour showed an improvement in the Se status [30,31], although no influence has been observed on biomarkers of cardiovascular disease risk, oxidative stress, and immune function [31]. The highest Se levels in Finland and Italy could just be related to the implementation of a nationwide Se fertilization program [32] or biofortification of potatoes, onions, or carrots [33], respectively.

However, circulating trace elements, including Se, are very responsive to aging and many other physio-pathological changes occurring with aging, independently of nutrition. Moreover, we did not perform a recruitment in order to be representative for the whole population in each country. Hence, further studies are needed before providing definitive population-tailored advice on Se biofortification. 

In the RASIG subjects, plasma Se levels were associated with age; after sex stratification, plasma Se decreased linearly with aging in males, while females showed a U-shape trend. Other investigations have reported a decrement in blood Se with age. Results from the EPIC-Potsdam cohort study show a decrease in serum Se concentrations in a German population of healthy septuagenarian subjects over a 20-year follow-up period [34]. Another study, in a Chinese population in the age range 65–102 years, shows an age-related decrease in blood Se concentrations [35]. In contrast to our work, however, those studies were focused on old or very old subjects and did not cover the younger age range in females, where we found that Se levels in our population were similar in younger females compared to those in the 65–75 age range. The apparent discrepancies may depend on the differences in environmental factors, lifestyle, and dietary habits, but also on the influence of the estrogen levels on Se metabolism and tissue distribution [36]. 

Regarding the effect of dietary habits, we found that higher fish and vitamin consumption is associated with increased plasma Se levels, as also supported by other investigations [22,37]. In fact, studies on dietary Se intake showed that fish represents one of the main contributors to Se dietary intake in European populations [38]. 

We found no significant differences in plasma Se concentrations between RASIG, GO, and SGO, while differences were found in Se fractionation between subject groups.

To our knowledge, this is the first study to investigate the Se fractionation in elderly subjects from long-lived families. We have found significant differences in the percentage of Se2, which was higher in RASIG compared to GO and SGO subjects, suggesting an effect of lifestyle and dietary habits on Se fractionation, whereas a minor influence seems to be exerted by genetics. The lack of differences between Se plasma levels and its fractions in GO and SGO participants is not surprising because, as previously observed in other studies within the MARK-AGE project, these parameters may depend on sharing a common environment, diet, and lifestyle factors [29,39,40]. Importantly, in this study, we were not able to identify all Se fractions coeluting with the three major peaks, and thus the contribution of minor species to the observed differences cannot be excluded. 

However, Se is known to be bound to plasma glutathione peroxidase (Se-GpX, 15–25%), selenoprotein P (50–60%), and albumin (10–20%) [41]. As the first peak of Se (Se1) coelutes with the big fraction of zinc, we can suppose that Se1 might include Se bound to albumin [23]. Given that SELENOP is the major selenoprotein in plasma and contains ten selenocysteine residues (compared to only four in GPx), it is conceivable that Se1a (the most abundant peak) corresponds to SELENOP, while Se1b corresponds to SeAlb. The positive association between Se1a and serum albumin levels (Appendix A) does not contradict this hypothesis, as a correlation between these two proteins has also been documented by others [42]. However, further investigations combining this separation with molecular mass spectrometry or other detection systems [43,44] will be required to precisely identify these Se species. 

Interestingly, the relative percentage of Se1a showed a linear increase with age in females, in agreement with a previous observation supporting an age-related increase in this protein [45]. The linear decrease in Se1b percentage with age in females could be in line with previous results demonstrating an inverse association between Se-dependent GPx and age in a cohort of 601 women older than 65 years of age [46], while the lack of association between Se fractions and age in males may be consistent with previous findings [22], especially after adjusting for confounding factors. 

Our results also support other evidence indicating an inverse relationship between plasma Se levels and BMI [47]. A positive association was also observed between plasma Se levels and total cholesterol, HDL, triglycerides, and fibrinogen, as also reported by other authors [48,49].

A recent study on a large Chinese cohort has shown that higher blood Se levels are associated with a significantly lower risk of prevalent hyperhomocysteinemia [50]. In line with those results, we observed a significant negative association between plasma Se and serum homocysteine. 

No significant association was found with inflammatory (CRP and ceruloplasmin) and anti-inflammatory (adiponectin) parameters, although some evidence was reported for increased inflammation in the presence of Se deficiency [5]. The lack of association between Se plasma concentration and glycemic biomarkers would seem contradictory with some studies where this relationship has been observed [51,52]. However, the results remain controversial [53], reporting sex differences [54] or association in diabetic patients, but not in healthy subjects [55].

Interestingly, Se2 fraction was negatively associated with fibrinogen and positively associated with adiponectin, H1Abc, and ceruloplasmin. In support of these findings, other studies report a negative association between fibrinogen and Se-dependent Gpx levels in patients with stroke or age-related macular degeneration [56,57] and a direct relationship between the overexpression of Se-dependent Gpx and the development of insulin resistance and diabetes in mice models [58,59]. 

In conclusion, these findings show that sex-specific, nutritional, and inflammatory factors play a crucial role in the regulation of Se plasma levels throughout the aging process and that the shared environment of GO and SGO plays a role in their distinctive Se fractionation. 

## Figures and Tables

**Figure 1 nutrients-15-02182-f001:**
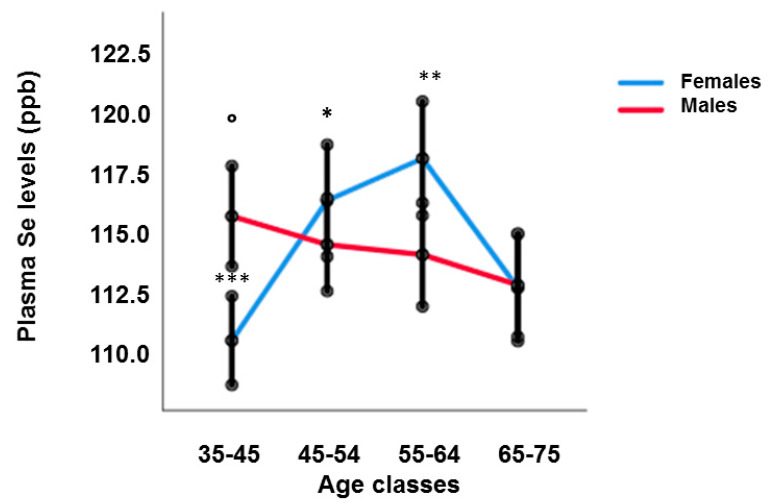
Plasma Se levels in RASIG population after sex stratification. Males over the age of 65 had lower Se plasma levels than subjects in the 35–44 age group (° *p* < 0.05), Females in the 35–44 age group had lower Se plasma levels than those in the 45–54 and 55–64 age groups (*** *p* < 0.0001), Females over 65 showed reduced Se plasma levels compared to those in the 45–54 (* *p* < 0.05) and 55–64 (** *p* = 0.001) age groups.

**Figure 2 nutrients-15-02182-f002:**
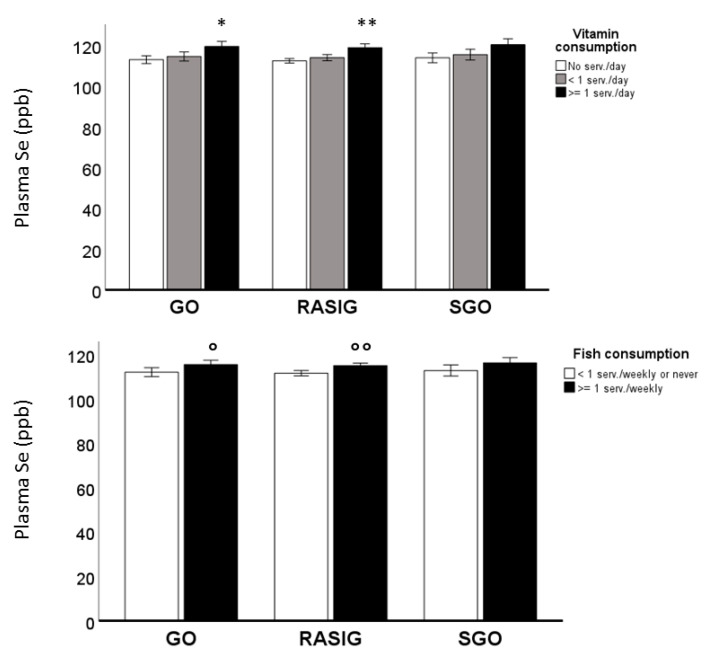
Plasma Se levels in MARK-AGE population in relation to vitamin and fish consumption. Plasma Se levels increased in RASIG and GO subjects who took more vitamins and consumed more fish. ** *p* < 0.0001 compared to no serv./day; <1 serv./day; * *p* < 0.01 compared to no serv./day ° *p* < 0.05 compared to <1 serv./weekly or never; °° *p* < 0.001 compared to <1 serv./weekly or never.

**Figure 3 nutrients-15-02182-f003:**
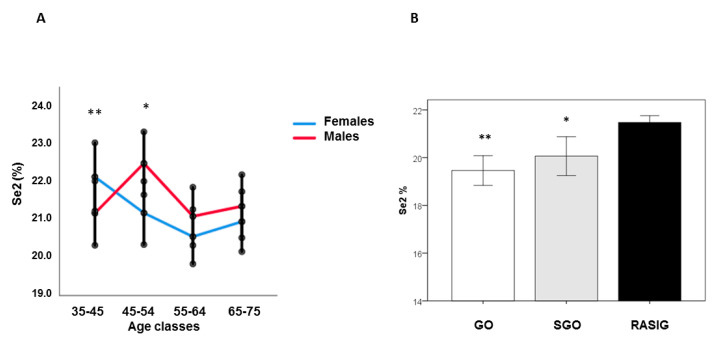
Se2 percentage in GO, SGO, and RASIG subjects in the whole sample and after sex stratification. Younger females had lower Se2 than those in the 55–64 and 65–75 age groups (** *p* < 0.01). Males in the 45–54 age group had higher Se2 than subjects in the 35–44 and 55–64 age groups (* *p* < 0.05, ((**A**) Relation of Se2 percentage with age after sex stratification in the whole sample). The percentage of Se2 fraction was higher in RASIG than GO and SGO ((**B**) Se2 percentage in GO, SGO and RASIG participants), * *p* < 0.05 compared to RASIG; ** *p* < 0.001 compared to RASIG.

**Table 1 nutrients-15-02182-t001:** Effect of age, gender, country, and BMI on plasma Se concentrations in the RASIG population.

		Stat	N	Plasma Se (ppb) Mean ± SD or (CI 95%)	*p* (GLM)
Age group (years)	35–44	a	481	112.09 ± 17.59 ^c^	<0.001
	45–54	b	552	114.71 ± 19.71	
	55–64	c	595	115.13 ± 23.49 ^a,d^	
	65–75	d	572	112.27 ± 20.73 ^c^	
Center	Finland	a	90	133.24 ± 18.55 ^c,d,e,g,h^	<0.001
	Italy	b	383	127.57 ± 12.14 ^c,d,e,g,h^	
	Austria	c	380	107.11 ± 18.11 ^a,b,e,g^	
	Greece	d	374	105.02 ± 14.68 ^a,b,e,g^	
	Poland	e	371	95.14 ± 17.26 ^a,b,c,d,g,h^	
	The Netherlands	f	-	-	
	Belgium	g	254	120.46 ± 25.47 ^a,b,c,d,g,h^	
	Germany	h	345	107.15 ± 15.66 ^a,b,e,g^	
Sex	F	a	1133	111.73 ± 20.79	0.57
	M	b	1064	110.05 ± 20.56	
BMI classes	<25	a	988	113.16 ± 20.08 ^c^	<0.001
	25 to <30	b	826	109.84 ± 21.53	
	≥30	c	378	107.41 ± 19.92 ^a^	

Data are reported as uncorrected means ± SD. Statistical comparison was performed with Generalized Linear Models (GLMs) using a normal distribution with identity-link function model. Bonferroni test was used as post-hoc for pairwise comparisons. Significant different groups are identified by superscripts indicated in the column “stats”. For the investigation of age-group effect, the model included the effects of sex and recruitment center as factors and BMI (continuous variable). For the investigation of BMI class effects, the model included the effects of sex and recruitment center as factors and age (continuous variable). All other GLM models included the effects of sex and recruitment center as well as age and BMI (continuous variables) as covariates.

**Table 2 nutrients-15-02182-t002:** Multivariate stepwise linear regression analysis for glucometabolic, inflammatory, and anti-inflammatory parameters independently associated with plasma Se levels and Se2 fraction in the RASIG population.

	Unstandardized Coefficients		Standardized Coefficients	
Plasma Se Levels	B	Std. Error	Beta	*p* Value
Albumin	1.301	0.144	0.197	*p* < 0.0001
HDL	3.917	1.243	0.083	0.002
TC	1.484	0.474	0.073	0.002
Fibrinogen	1.048	0.320	0.071	0.001
Homocysteine	−0.200	0.078	−0.056	0.010
TG	1.155	0.573	0.048	0.044
**Se2 fraction**	**B**	**Std. Error**	**Beta**	***p* value**
Fibrinogen	−0.596	0.115	−0.123	*p* < 0.0001
Adiponectin	0.112	0.023	0.124	*p* < 0.0001
HbA1c	0.854	0.251	0.077	0.001
Ceruloplasmin	0.064	0.023	0.068	0.005

TC: total cholesterol, HDL: HDL cholesterol, TG: triglycerides, HbA1c: glycated hemoglobin.

## Data Availability

The datasets analyzed in this study are available from the corresponding authors upon reasonable request.

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
