# Peer review of "Uncovering the Relationship between Selenium Status, Age, Health, and Dietary Habits: Insights from a Large Population Study including Nonagenarian Offspring from the MARK-AGE Project"

_nutrients, 2023, doi:10.3390/nu15092182_

Round 1

Reviewer 1 Report

This is very interesting topic, well designed study and written paper. Few questions arrise, based upon difference between subgroups involved in the study. Would authors reccomend additional Se supplementation based on different countries from which examinees originate ( Se biofortification) and based on their age groups? 

Author Response

This is very interesting topic, well designed study and written paper. Few questions arrise, based upon difference between subgroups involved in the study. Would authors recommend additional Se supplementation based on different countries from which examinees originate ( Se biofortification) and based on their age groups? 

We thanks the reviewer for his comment. We have integrated the discussion with the following part:” The relatively low levels of Se observed in the other countries (Greece, Poland and Austria) might suggest the use of Se biofortification to produce Se-enriched foods for counteracting Se deficiency. Consumption of Se-biofortified wheat or maize flour show an improvement in the Se status [30,31], although no influence has been observed on biomarkers of cardi-ovascular disease risk, oxidative stress, and immune function [31]. The highest Se levels in Finland and Italy could just be related to the implementation of nationwide Se fertilization programme [32] or biofortification of potatoes, onions or carrots [33], respectively.

However, circulating trace elements including Se are very responsive to aging and many other physio-pathological changes occurring with aging independently by nutri-tion. Moreover, we did not perform a recruitment in order to be representative for the whole population in each country. Hence, further studies are needed before providing de-finitive population tailored advices on Se biofortification.”

Reviewer 2 Report

The paper of Giacconi et al. is an interesting epidemiological survey on Se in longevity. The study is properly designed, large in sample size and uses modern analytical technology. I find it highly relevant for the journal. However, certain important points should be clarified before further consideration.

Major comments:

1) I may question the use of term speciation in the current work since the peaks were not appropriately identified. I think "fractionation" is the better term according to the accepted definitions. See the definition paper by Templeton et al. (doi: 10.1351/pac200072081453) for clarity. Also, in line with this, please present a sample chromatogram (in the supplement) and explain the validation of the technique in brief in the methods (or in the supplement too).

2) Use modern selenoprotein nomenclature (see Gladyshev et al. doi:  10.1074/jbc.M116.756155). SeP is an outdated acronym. 

Minor comments

Lines 47, 49 and 57 A space is missing after the period.

Line 199 Do you mean slope here? It is unclear (also check figure captions)

Lines 318-323 This is not discussion (rather intro) I suggest removing this paragraph since similar things were already mentioned in the intro.

I suggest presenting the supplement as PDF at revision.

Author Response

1) I may question the use of term speciation in the current work since the peaks were not appropriately identified. I think "fractionation" is the better term according to the accepted definitions. See the definition paper by Templeton et al. (doi: 10.1351/pac200072081453) for clarity. Also, in line with this, please present a sample chromatogram (in the supplement) and explain the validation of the technique in brief in the methods (or in the supplement too).

We thank the reviewer for this suggestion. According to the Guidelines for terms related to chemical speciation and fractionation of elements, by Templeton et al., we replaced the term speciation with fractionation and Se species with Se fractions. Concerning the Methods section, we improved the fractionation analysis description by adding several information reported in the previously published manuscript (ref.23) in which the method was validated and largely explained. In this previous article, we validated the quantification of the peaks by two different methods (in the manuscript defined PAN and CC). Considering that in this manuscript, we only included Se percentages, we improved the methods section by adding technical explanations on the retention times used to define Se1 and Se2.

Considering that in the previous manuscript (ref. 23) it was already published a sample chromatogram of Se “speciation” (Figure 1), following the reviewer suggestion, a sample chromatogram representative of the percentage differences between GO and RASIG was included in the Supplementary data of this manuscript (Suppl. Figure 6).

Ref. 23

-           Malavolta M, Piacenza F, Basso A, Giacconi R, Costarelli L, Pierpaoli S, Mocchegiani E. Speciation of trace elements in human serum by micro anion exchange chromatography coupled with inductively coupled plasma mass spectrometry. Anal Biochem. 2012 Feb 1;421(1):16-25. doi: 10.1016/j.ab.2011.11.004.

2) Use modern selenoprotein nomenclature (see Gladyshev et al. doi:  10.1074/jbc.M116.756155). SeP is an outdated acronym.

We thanks the reviewer for his suggestion. SeP has been replaced with the new acronym SELENOP.

Minor comments

-Lines 47, 49 and 57 A space is missing after the period.

The space has been inserted.

-Line 199 Do you mean slope here? It is unclear (also check figure captions)

The Supplementary figure 1 caption has been changed as follow: Linear regression between Plasma Se  and age (slope 0.003; p=0.878). Quadratic regression between Plasma Se  and age (costant=65.1, b1=1.958, b2=-0.018; p<0.001)

- Lines 318-323 This is not discussion (rather intro) I suggest removing this paragraph since similar things were already mentioned in the intro.

As suggested by the reviewer the sentences have been removed and partially integrated in the introduction section.

-I suggest presenting the supplement as PDF at revision.

The supplementary material has been replaced with a pdf file

Round 2

Reviewer 2 Report

I thank the authors for adequately addressing my comments